# Memoized Online Variational Inference for Dirichlet Process Mixture Models

**Michael C. Hughes and Erik B. Sudderth**
Department of Computer Science, Brown University, Providence, RI 02912
`mhughes@cs.brown.edu, sudderth@cs.brown.edu`

## Abstract

Variational inference algorithms provide the most effective framework for large-scale training of Bayesian nonparametric models. Stochastic online approaches are promising, but are sensitive to the chosen learning rate and often converge to poor local optima. We present a new algorithm, *memoized online variational inference*, which scales to very large (yet finite) datasets while avoiding the complexities of stochastic gradient. Our algorithm maintains finite-dimensional sufficient statistics from batches of the full dataset, requiring some additional memory but still scaling to millions of examples. Exploiting nested families of variational bounds for infinite nonparametric models, we develop principled birth and merge moves allowing non-local optimization. Births adaptively add components to the model to escape local optima, while merges remove redundancy and improve speed. Using Dirichlet process mixture models for image clustering and denoising, we demonstrate major improvements in robustness and accuracy.

## 1 Introduction

Bayesian nonparametric methods provide a flexible framework for unsupervised modeling of structured data like text documents, time series, and images. They are especially promising for large datasets, as their nonparametric priors should allow complexity to grow smoothly as more data is seen. Unfortunately, contemporary inference algorithms do not live up to this promise, scaling poorly and yielding solutions that represent poor local optima of the true posterior. In this paper, we propose new scalable algorithms capable of escaping local optima. Our focus is on clustering data via the Dirichlet process (DP) mixture model, but our methods are much more widely applicable.

Stochastic online variational inference is a promising general-purpose approach to Bayesian non-parametric learning from streaming data [1]. While individual steps of stochastic optimization algorithms are by design scalable, they are extremely vulnerable to local optima for non-convex unsupervised learning problems, frequently yielding poor solutions (see Fig. 2). While taking the best of multiple runs is possible, this is unreliable, expensive, and ineffective in more complex structured models. Furthermore, the noisy gradient step size (or learning rate) requires external parameters which must be fine-tuned for best performance, often requiring an expensive validation procedure. Recent work has proposed methods for automatically adapting learning rates [2], but these algorithms' progress on the overall variational objective remains local and non-monotonic.

In this paper, we present an alternative algorithm, *memoized online variational inference*, which avoids noisy gradient steps and learning rates altogether. Our method is useful when all data may not fit in memory, but we can afford multiple full passes through the data by processing successive batches. The algorithm visits each batch in turn and updates a cached set of sufficient statistics which accurately reflect the *entire dataset*. This allows rapid and noise-free updates to global parameters at every step, quickly propagating information and speeding convergence. Our memoized approach is generally applicable in any case batch or stochastic online methods are useful, including topic models [1] and relational models [3], though we do not explore these here.

We further develop a principled framework for escaping local optima in the online setting, by integrating birth and merge moves within our algorithm's coordinate ascent steps. Most existing mean-field algorithms impose a restrictive fixed truncation in the number of components, which is hard to set *a priori* on big datasets: either it is too small and inexpressive, or too large and computationally inefficient. Our birth and merge moves, together with a *nested* variational approximation to the posterior, enable adaptive creation and pruning of clusters on-the-fly. Because these moves are validated by an exactly tracked global variational objective, we avoid potential instabilities of stochastic online split-merge proposals [4]. The structure of our moves is very different from split-merge MCMC methods [5, 6]; applications of these algorithms have been limited to hundreds of data points, while our experiments show scaling of memoized split-merge proposals to millions of examples.

We review the Dirichlet process mixture model and variational inference in Sec. 2, outline our novel memoized algorithm in Sec. 3, and evaluate on clustering and denoising applications in Sec. 4.

## 2 Variational inference for Dirichlet process mixture models

The *Dirichlet process* (DP) provides a nonparametric prior for partitioning exchangeable datasets into discrete clusters [7]. An instantiation $G$ of a DP is an infinite collection of atoms, each of which represents one mixture component. Component $k$ has mixture weight $w_k$ sampled as follows:

$$G \sim \text{DP}(\alpha_0 H), \quad G \triangleq \sum_{k=1}^{\infty} w_k \delta_{\phi_k}, \quad v_k \sim \text{Beta}(1, \alpha_0), \quad w_k = v_k \prod_{\ell=1}^{k-1} (1 - v_\ell). \quad (1)$$

This *stick-breaking* process provides mixture weights and parameters. Each data item $n$ chooses an assignment $z_n \sim \text{Cat}(w)$, and then draws observations $x_n \sim F(\phi_{z_n})$. The data-generating parameter $\phi_k$ is drawn from a base measure $H$ with natural parameters $\lambda_0$. We assume both $H$ and $F$ belong to exponential families with log-normalizers $a$ and sufficient statistics $t$:

$$p(\phi_k \mid \lambda_0) = \exp\left\{\lambda_0^T t_0(\phi_k) - a_0(\lambda_0)\right\}, \qquad p(x_n \mid \phi_k) = \exp\left\{\phi_k^T t(x_n) - a(\phi_k)\right\}. \quad (2)$$

For simplicity, we assume unit reference measures. The goal of inference is to recover stick-breaking proportions $v_k$ and data-generating parameters $\phi_k$ for each global mixture component $k$, as well as discrete cluster assignments $z = \{z_n\}_{n=1}^N$ for each observation. The joint distribution is

$$p(\mathbf{x}, \mathbf{z}, \phi, v) = \prod_{n=1}^{N} F(x_n \mid \phi_{z_n}) \text{Cat}(z_n \mid w(v)) \prod_{k=1}^{\infty} \text{Beta}(v_k \mid 1, \alpha_0) H(\phi_k \mid \lambda_0) \quad (3)$$

While our algorithms are directly applicable to any DP mixture of exponential families, our experiments focus on $D$-dimensional real-valued data $x_n$, for which we take $F$ to be Gaussian. For some data, we consider full-mean, full-covariance analysis (where $H$ is normal-Wishart), while other applications consider zero-mean, full-covariance analysis (where $H$ is Wishart).

### 2.1 Mean-field variational inference for DP mixture models

To approximate the full (but intractable) posterior over variables $z, v, \phi$, we consider a fully-factorized variational distribution $q$, with individual factors from appropriate exponential families:[1]

$$q(\mathbf{z}, v, \phi) = \prod_{n=1}^{N} q(z_n | \hat{r}_n) \prod_{k=1}^{K} q(v_k | \hat{\alpha}_1, \hat{\alpha}_0) q(\phi_k | \hat{\lambda}_k), \quad (4)$$

$$q(z_n) = \text{Cat}(z_n \mid \hat{r}_{n1}, \dots \hat{r}_{nK}), \quad q(v_k) = \text{Beta}(v_k \mid \hat{\alpha}_{k1}, \hat{\alpha}_{k0}), \quad q(\phi_k) = H(\phi_k \mid \hat{\lambda}_k). \quad (5)$$

To tractably handle the infinite set of components available under the DP prior, we truncate the discrete assignment factor to enforce $q(z_n = k) = 0$ for $k > K$. This forces all data to be explained by only the first $K$ components, inducing *conditional independence* between observed data and any global parameters $v_k, \phi_k$ with index $k > K$. Inference may thus focus exclusively on a finite set of $K$ components, while reasonably approximating the true infinite posterior for large $K$.

Crucially, our truncation is *nested*: any learned $q$ with truncation $K$ can be represented exactly under truncation $K+1$ by setting the final component to have zero mass. This truncation, previously advocated by [8, 4], has considerable advantages over non-nested direct truncation of the stick-breaking process [7], which places artifically large mass on the final component. It is more efficient and broadly applicable than an alternative trunction which sets the stick-breaking "tail" to its prior [9].

Variational algorithms optimize the parameters of $q$ to minimize the KL divergence from the true, intractable posterior [7]. The optimal $q$ maximizes the *evidence lower bound* (ELBO) objective $\mathcal{L}$:

$$\log p(\mathbf{x} \mid \alpha_0, \lambda_0) \geq \mathcal{L}(q) \triangleq \mathbb{E}_q\Big[ \log p(\mathbf{x}, v, \mathbf{z}, \phi \mid \alpha_0, \lambda_0) - \log q(v, \mathbf{z}, \phi) \Big] \qquad (6)$$

For DP mixtures of exponential family distributions, $\mathcal{L}(q)$ has a simple form. For each component $k$, we store its expected mass $\hat{N}_k$ and expected sufficient statistic $s_k(\mathbf{x})$. All but one term in $\mathcal{L}(q)$ can then be written using only these summaries and expectations of the global parameters $v, \phi$:

$$\hat{N}_k \triangleq \mathbb{E}_q\Big[ \sum_{n=1}^N z_{nk} \Big] = \sum_{n=1}^N \hat{r}_{nk}, \qquad s_k(\mathbf{x}) \triangleq \mathbb{E}_q\Big[ \sum_{n=1}^N z_{nk} t(x_n) \Big] = \sum_{n=1}^N \hat{r}_{nk} t(x_n), \qquad (7)$$

$$\mathcal{L}(q) = \sum_{k=1}^K \Bigg( \mathbb{E}_q[\phi_k]^T s_k(\mathbf{x}) - \hat{N}_k \mathbb{E}_q[a(\phi_k)] + \hat{N}_k \mathbb{E}_q[\log w_k(v)] - \sum_{n=1}^N \hat{r}_{nk} \log \hat{r}_{nk}$$
$$+ \mathbb{E}_q\bigg[ \log \frac{\mathrm{Beta}(v_k \mid 1, \alpha_0)}{q(v_k \mid \hat{\alpha}_{k1}, \hat{\alpha}_{k0})} \bigg] + \mathbb{E}_q\bigg[ \log \frac{H(\phi_k \mid \lambda_0)}{q(\phi_k \mid \hat{\lambda}_k)} \bigg] \Bigg) \qquad (8)$$

Excluding the entropy term $-\sum \hat{r}_{nk} \log \hat{r}_{nk}$ which we discuss later, this bound is a simple linear function of the summaries $\hat{N}_k, s_k(\mathbf{x})$. Given precomputed entropies and summaries, evaluation of $\mathcal{L}(q)$ can be done in time *independent* of the data size $N$. We next review variational algorithms for optimizing $q$ via coordinate ascent, iteratively updating individual factors of $q$. We describe algorithms in terms of two updates [1]: *global* parameters (stick-breaking proportions $v_k$ and data-generating parameters $\phi_k$), and *local* parameters (assignments of data to components $z_n$).

## 2.2 Full-dataset variational inference

Standard full-dataset variational inference [7] updates local factors $q(z_n \mid \hat{r}_n)$ for *all* observations $n = 1, \ldots, N$ by visiting each item $n$ and computing the fraction $\hat{r}_{nk}$ explained by component $k$:

$$\tilde{r}_{nk} = \exp\Big( \mathbb{E}_q[\log w_k(v)] + \mathbb{E}_q[\log p(x_n \mid \phi_k)] \Big), \quad \hat{r}_{nk} = \frac{\tilde{r}_{nk}}{\sum_{\ell=1}^K \tilde{r}_{n\ell}}. \qquad (9)$$

Next, we update global factors $q(v_k|\hat{\alpha}_{k1}, \hat{\alpha}_{k0}), q(\phi_k|\hat{\lambda}_k)$ for each component $k$. After computing summary statistics $\hat{N}_k, s_k(\mathbf{x})$ given the new $\hat{r}_{nk}$ via Eq. (7), the update equations become

$$\hat{\alpha}_{k1} = \alpha_1 + \hat{N}_k, \qquad \hat{\alpha}_{k0} = \alpha_0 + \sum_{\ell=k+1}^K \hat{N}_\ell, \qquad \hat{\lambda}_k = \lambda_0 + s_k(\mathbf{x}). \qquad (10)$$

While simple and guaranteed to converge, this approach scales poorly to big datasets. Because global parameters are updated only after a full pass through the data, information propagates slowly.

## 2.3 Stochastic online variational inference

Stochastic online (SO) variational inference scales to huge datasets [1]. Instead of analyzing all data at once, SO processes only a subset ("batch") $\mathcal{B}_t$ at each iteration $t$. These subsets are assumed sampled uniformly at random from a larger (but *fixed* size $N$) corpus. Given a batch, SO first updates local factors $q(z_n)$ for $n \in \mathcal{B}_t$ via Eq. (9). It then updates global factors via a *noisy gradient* step, using sufficient statistics of $q(z_n)$ from only the current batch. These steps optimize a noisy function, which in expectation (with respect to batch sampling) converges to the true objective (6).

*Natural* gradient steps are computationally tractable for exponential family models, involving nearly the same computations as the full-dataset updates [1]. For example, to update the variational parameter $\hat{\lambda}_k$ from (5) at iteration $t$, we first compute the global update given only data in the current batch,

amplified to be at full-dataset scale: $\hat{\lambda}_k^* = \lambda_0 + \frac{N}{|\mathcal{B}_t|} s_k(\mathcal{B}_t)$. Then, we interpolate between this and the previous global parameters to arrive at the final result: $\hat{\lambda}_k^{(t)} \leftarrow \rho_t \hat{\lambda}_k^* + (1 - \rho_t) \hat{\lambda}_k^{(t-1)}$. The learning rate $\rho_t$ controls how "forgetful" the algorithm is of previous values; if it decays at appropriate rates, stochastic inference provably converges to a *local* optimum of the global objective $\mathcal{L}(q)$ [1].

This online approach has clear computational advantages and can sometimes yield higher quality solutions than the full-data algorithm, since it conveys information between local and global parameters more frequently. However, performance is extremely sensitive to the learning rate decay schedule and choice of batch size, as we demonstrate in later experiments.

## 3  Memoized online variational inference

Generalizing previous incremental variants of the *expectation maximization* (EM) algorithm [10], we now develop our *memoized online variational inference* algorithm. We divide the data into $B$ fixed batches $\{\mathcal{B}_b\}_{b=1}^{B}$. For each batch, we maintain *memoized* sufficient statistics $S_k^b = [\hat{N}_k(\mathcal{B}_b), s_k(\mathcal{B}_b)]$ for each component $k$. We also track the full-dataset statistics $S_k^0 = [\hat{N}_k, s_k(\mathbf{x})]$. These compact summary statistics allow guarantees of correct full-dataset analysis while processing only one small batch at a time. Our approach hinges on the fact that these sufficient statistics are *additive*: summaries of an entire dataset can be written exactly as the addition of summaries of distinct batches. Note that our memoization of deterministic analyses of batches of data is distinct from the stochastic memoization, or "lazy" instantiation, of random variables in some Monte Carlo methods [11, 12].

Memoized inference proceeds by visiting (in random order) each distinct batch once in a full pass through the data, incrementally updating the local and global parameters related to that batch $b$. First, we update local parameters for the current batch ($q(z_n \mid \hat{r}_n)$ for $n \in \mathcal{B}_b$) via Eq. (9). Next, we update cached global sufficient statistics for each component: we subtract the old (cached) summary of batch $b$, compute a new batch-level summary, and add the result to the full-dataset summary:

$$S_k^0 \leftarrow S_k^0 - S_k^b, \quad S_k^b \leftarrow \Big[ \sum_{n \in \mathcal{B}_b} \hat{r}_{nk}, \sum_{n \in \mathcal{B}_b} \hat{r}_{nk} t(x_n) \Big], \quad S_k^0 \leftarrow S_k^0 + S_k^b. \tag{11}$$

Finally, given the new full-dataset summary $S_k^0$, we update global parameters exactly as in Eq. (10). Unlike stochastic online algorithms, memoized inference is guaranteed to improve the full-dataset ELBO at every step. Correctness follows immediately from the arguments in [10]. By construction, each local or global step will result in a new $q$ that strictly increases the objective $\mathcal{L}(q)$.

In the limit where $B = 1$, memoized inference reduces to standard full-dataset updates. However, given many batches it is far more scalable, while maintaining all guarantees of batch inference. Furthermore, it generally converges faster than the full-dataset algorithm due to frequently interleaving global and local updates. Provided we can store memoized sufficient statistics for each batch (*not* each observation), memoized inference has the same computational complexity as stochastic methods while avoiding noise and sensitivity to learning rates. Recent analysis of convex optimization algorithms [13] demonstrated theoretical and practical advantages for methods that use cached full-dataset summaries to update parameters, as we do, instead of stochastic current-batch-only updates.

This memoized algorithm can compute the full-dataset objective $\mathcal{L}(q)$ exactly at any point (after visiting all items once). To do so efficiently, we need to compute and store the assignment entropy $H_k^b = -\sum_{n \in \mathcal{B}_b} r_{nk} \log r_{nk}$ after visiting each batch $b$. We also need to track the full-data entropy $H_k^0 = \sum_{b=1}^{B} H_k^b$, which is additive just like the sufficient statistics and incrementally updated after each batch. Given both $H_k^0$ and $S_k^0$, evaluation of the full-dataset ELBO in Eq. (8) is exact and rapid.

### 3.1  Birth moves to escape local optima

We now propose additional *birth* moves that, when interleaved with conventional coordinate ascent parameter updates, can add useful new components to the model and escape local optima. Previous methods [14, 9, 4] for changing variational truncations create just one extra component via a "split" move that is highly-specialized to particular likelihoods. Wang and Blei [15] explore truncation levels via a local collapsed Gibbs sampler, but samplers are slow to make large changes. In contrast, our births add *many* components at once and apply to *any* exponential family mixture model.

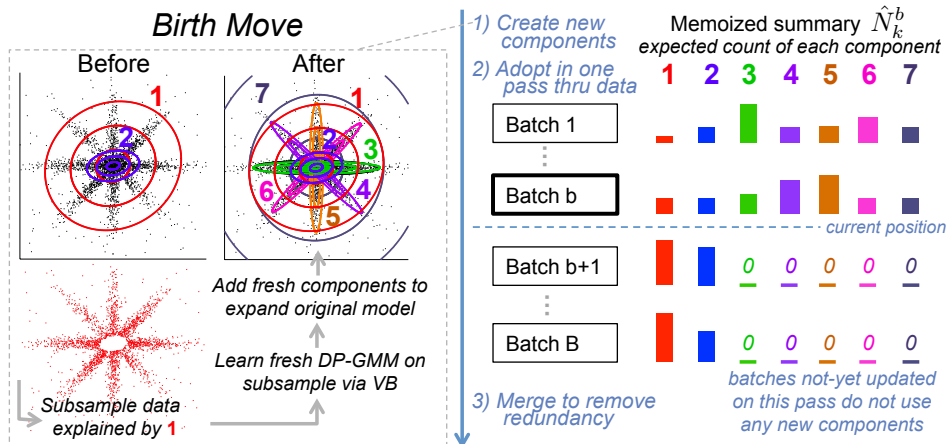

Figure 1: One pass through a toy dataset for memoized learning with birth and merge moves (MO-BM), showing creation (left) and adoption (right) of new components. *Left:* Scatter plot of 2D observed data, and a subsample targeted via the first mixture component. Elliptical contours show component covariance matrices. *Right:* Bar plots of memoized counts $\hat{N}_k^b$ for each batch. *Not shown:* Memoized sufficient statistics $s_k^b$.

Creating new components in the online setting is challenging. Each small batch may not have enough examples of a missing component to inspire a good proposal, even if that component is well-supported by the full dataset. We thus advocate birth moves that happen in three phases (collection, creation, and adoption) over two passes of the data. The first pass collects a targeted data sample more likely to yield informative proposals than a small, predefined batch. The second pass, shown in Fig. 1, creates new components and then updates every batch with the expanded model. Successive births are interleaved; at each pass there are proposals both active and in preparation. We sketch out each step of the algorithm below. For complete details, see the supplement.

**Collection**     During pass 1, we collect a *targeted* subsample $\mathbf{x}'$ of the data, of size at most $N' = 10^4$. This subsample targets a single component $k'$. When visiting each batch, we copy data $x_n$ into $\mathbf{x}'$ if $\hat{r}_{nk'} > \tau$ (we set $\tau = 0.1$). This threshold test ensures the subsample contains related data, but also promotes diversity by considering data explained partially by other components $k \neq k'$. Targeted samples vary from iteration to iteration because batches are visited in distinct, random orders.

**Creation**     Before pass 2, we *create* new components by fitting a DP mixture model with $K'$ (we take $K' = 10$) components to $\mathbf{x}'$, running variational inference for a limited budget of iterations. Taking advantage of our nested truncation, we expand our current model to include all $K + K'$ components, as shown in Fig. 1. Unlike previous work [9, 4], we do not immediately assess the change in ELBO produced by these new components, and always accept them. We rely on subsequent merge moves (Sec. 3.2) to remove unneeded components.

**Adoption**     During pass 2, we visit each batch and perform local and global parameter updates for the expanded $(K + K')$-component mixture. These updates use expanded global summaries $S^0$ that include summaries $S'$ from the targeted analysis of $\mathbf{x}'$. This results in *two* interpretations of the subset $\mathbf{x}'$: assignment to original components (mostly $k'$) and assignment to brand-new components. If the new components are favored, they will gain mass via new assignments made at each batch. After the pass, we subtract away $S'$ to yield both $S^0$ and global parameters exactly consistent with the data $\mathbf{x}$. Any nearly-empty new component will likely be pruned away by later merges.

By adding *many* components at once, our birth move allows rapid escape from poor local optima. Alone, births may sometimes cause a slight ELBO decrease by adding unnecessary components. However, in practice merge moves reliably reject poor births and restore original configurations. In Sec. 4, births are so effective that runs started at $K = 1$ recover necessary components on-the-fly.

## 3.2  Merge moves that optimize the full data objective

The computational cost of inference grows with the number of components $K$. To keep $K$ small, we develop merge moves that replace two components with a single merged one. Merge moves were first explored for batch variational methods [16, 14]. For hierarchical DP topic models, stochastic

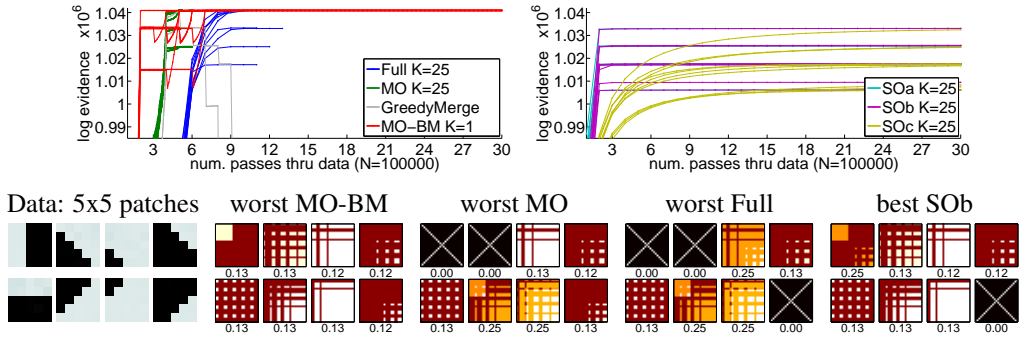

Figure 2: Comparison of full-data, stochastic (SO), and memoized (MO) on toy data with $K = 8$ true components (Sec. 4.1). *Top:* Trace of ELBO during training across 10 runs. SO compared with learning rates a,b,c. *Bottom Left:* Example patch generated by each component. *Bottom:* Covariance matrix and weights $w_k$ found by one run of each method, aligned to true components. "X": no comparable component found.

variational inference methods have been augmented to evaluate merge proposals based on noisy, single-batch estimates of the ELBO [4]. This can result in accepted merges that *decrease* the full-data objective (see Sec. 4.1 for an empirical illustration). In contrast, our algorithm accurately computes the full ELBO for each merge proposal, ensuring only useful merges are accepted.

Given two components $k_a, k_b$ to merge, we form a candidate $q'$ with $K - 1$ components, where merged component $k_m$ takes over all assignments to $k_a, k_b$: $\hat{r}_{nk_m} = \hat{r}_{nk_a} + \hat{r}_{nk_b}$. Instead of computing new assignments explicitly, additivity allows direct construction of merged global sufficient statistics: $S_{k_m}^0 = S_{k_a}^0 + S_{k_b}^0$. Merged global parameters follow from Eq. (10).

Our merge move has three steps: select components, form the candidate configuration $q'$, and accept $q'$ if the ELBO improves. Selecting $k_a, k_b$ to merge at random is unlikely to yield an improved configuration. After choosing $k_a$ at random, we select $k_b$ using a ratio of marginal likelihoods $M$ which compares the merged and separated configurations, easily computed with cached summaries:

$$p(k_b \mid k_a) \propto \frac{M(S_{k_a} + S_{k_b})}{M(S_{k_a})M(S_{k_b})}, \quad M(S_k) = \exp\left(a_0(\lambda_0 + s_k(\mathbf{x}))\right). \quad (12)$$

Our memoized approach allows *exact* evaluation of the full-data ELBO to compare the existing $q$ to merge candidate $q'$. As shown in Eq. (8), evaluating $\mathcal{L}(q')$ is a linear function of merged sufficient statistics, except for the assignment entropy term: $H_{ab} = -\sum_{n=1}^{N}(\hat{r}_{nk_a} + \hat{r}_{nk_b})\log(\hat{r}_{nk_a} + \hat{r}_{nk_b})$. We compute this term in advance for *all* possible merge pairs. This requires storing one set of $K(K - 1)/2$ scalars, one per candidate pair, for each batch. This modest precomputation allows rapid and exact merge moves, which improve model quality *and* speed-up post-merge iterations.

In one pass of the data, our algorithm performs a birth, memoized ascent steps for all batches, and several merges after the final batch. After a few passes, it recovers high-quality, compact structure.

## 4  Experimental results

We now compare algorithms for learning DP-Gaussian mixture models (DP-GMM), using our own implementations of full-dataset, stochastic online (SO), and memoized online (MO) inference, as well as our new birth-merge memoized algorithm (MO-BM). Code is available online. To examine SO's sensitivity to learning rate, we use a recommended [1] decay schedule $\rho_t = (t + d)^{-\kappa}$ with three diverse settings: a) $\kappa = 0.5, d = 10$, b) $\kappa = 0.5, d = 100$, and c) $\kappa = 0.9, d = 10$.

### 4.1  Toy data: How reliably do algorithms escape local optima?

We first study $N = 100000$ synthetic image patches generated by a zero-mean GMM with 8 equally-common components. Each one is defined by a $25 \times 25$ covariance matrix producing $5 \times 5$ patches with a strong edge. We investigate whether algorithms recover the true $K = 8$ structure. Each fixed-truncation method runs from 10 fixed random initializations with $K = 25$, while MO-BM starts at $K = 1$. Online methods traverse 100 batches (1000 examples per batch).

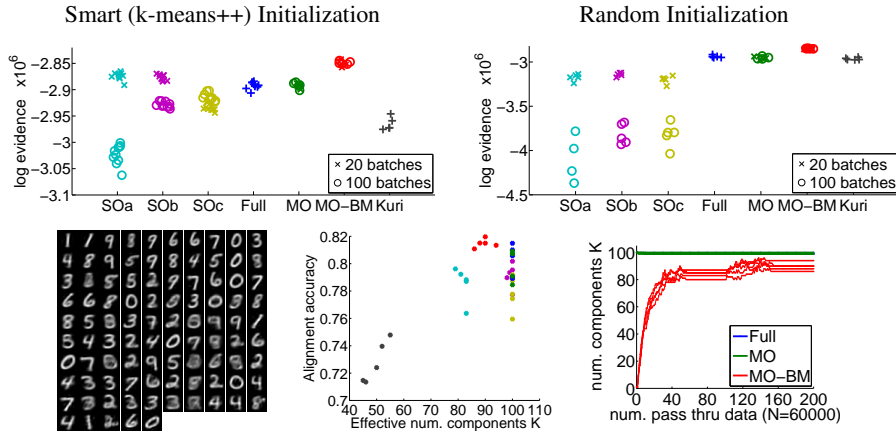

Figure 3: MNIST. *Top:* Comparison of final ELBO for multiple runs of each method, varying initialization and number of batches. Stochastic online (SO) compared at learning rates a,b,c. *Bottom left:* Visualization of cluster means for MO-BM's best run. *Bottom center:* Evaluation of cluster alignment to true digit label. *Bottom right:* Growth in truncation-level $K$ as more data visited with MO-BM.

Fig. 2 traces the training-set ELBO as more data arrives for each algorithm and shows estimated covariance matrices for the top 8 components for select runs. Even the best runs of SO do not recover ideal structure. In contrast, all 10 runs of our birth-merge algorithm find all 8 components, despite initialization at $K = 1$. The ELBO trace plots show this method escaping local optima, with slight drops indicating addition of new components followed by rapid increases as these are adopted. They further suggest that our fixed-truncation memoized method competes favorably with full-data inference, often converging to similar or better solutions after fewer passes through the data.

The fact that our MO-BM algorithm only performs merges that improve the full-data ELBO is crucial. Fig. 2 shows trace plots of GreedyMerge, a memoized online variant that instead uses only the current-batch ELBO to assess a proposed merge, as done in [4]. Given small batches (1000 examples each), there is not always enough data to warrant many distinct $25 \times 25$ covariance components. Thus, this method favors merges that in fact remove vital structure. All 5 runs of this GreedyMerge algorithm ruinously accept merges that decrease the full objective, consistently collapsing down to just one component. Our memoized approach ensures merges are always globally beneficial.

## 4.2 MNIST digit clustering

We now compare algorithms for clustering $N = 60000$ MNIST images of handwritten digits 0-9. We preprocess as in [9], projecting each image down to $D = 50$ dimensions via PCA. Here, we also compare to Kurihara's public implementation of variational inference with split moves [9]. MO-BM and Kurihara start at $K = 1$, while other methods are given 10 runs from two $K = 100$ initialization routines: random and smart (based on k-means++ [17]). For online methods, we compare 20 and 100 batches, and three learning rates. All runs complete 200 passes through the full dataset.

The final ELBO values for every run of each method are shown in Fig. 3. SO's performance varies dramatically across initialization, learning rate, and number of batches. Under random initialization, SO reaches especially poor local optima (note lower y-axis scale). In contrast, our memoized approach consistently delivers solutions on par with full inference, with no apparent sensitivity to the number of batches. With births and merges enabled, MO-BM expands from $K = 1$ to over 80 components, finding better solutions than every smart $K = 100$ initialization. MO-BM even outperforms Kurihara's offline split algorithm, yielding 30-40 more components and higher ELBO values. Altogether, Fig. 3 exposes SO's extreme sensitivity, validates MO as a more reliable alternative, and shows that our birth-merge algorithm is more effective at avoiding local optima.

Fig. 3 also shows cluster means learned by the best MO-BM run, covering many styles of each digit. We further compute a hard segmentation of the data using the $q(z)$ from smart initialization runs. Each DP-GMM cluster is aligned to one digit by majority vote of its members. A plot of alignment accuracy in Fig. 3 shows our MO-BM consistently among the best, with SO lagging significantly.

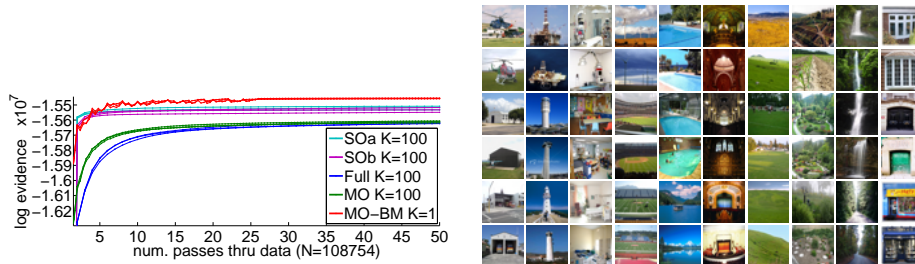

Figure 4: SUN-397 tiny images. *Left:* ELBO during training. *Right:* Visualization of 10 of 28 learned clusters for best MO-BM run. Each column shows two images from the top 3 categories aligned to one cluster.

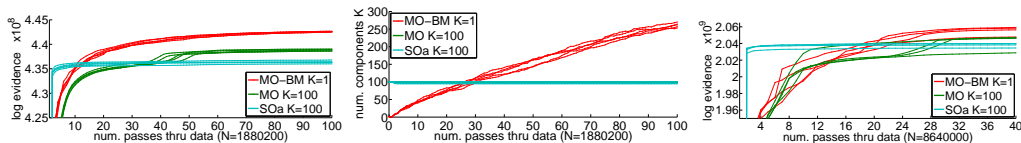

Figure 5: $8 \times 8$ image patches. *Left:* ELBO during training, $N = 1.88$ million. *Center:* Effective truncation-level $K$ during training, $N = 1.88$ million. *Right:* ELBO during training, $N = 8.64$ million.

## 4.3 Tiny image clustering

We next learn a full-mean DP-GMM for tiny, $32 \times 32$ images from the SUN-397 scene categories dataset [18]. We preprocess all $108754$ color images via PCA, projecting each example down to $D = 50$ dimensions. We start MO-BM at $K = 1$, while other methods have fixed $K = 100$. Fig. 4 plots the training ELBO as more data is seen. Our MO-BM runs surpass all other algorithms.

To verify quality, Fig. 4 shows images from the 3 most-related scene categories for each of several clusters found by MO-BM. For each learned cluster $k$, we rank all 397 categories to find those with the largest fraction of members assigned to $k$ via $\hat{r}_{\cdot k}$. The result is quite sensible, with clusters for tall free-standing objects, swimming pools and lakes, doorways, and waterfalls.

## 4.4 Image patch modeling

Our last experiment applies a zero-mean, full-covariance DP-GMM to learn the covariance structures of natural image patches, inspired by [19, 20]. We compare online algorithms on $N = 1.88$ million $8 \times 8$ patches, a dense subsampling of all patches from 200 images of the Berkeley Segmentation dataset. Fig. 5 shows that our birth-merge memoized algorithm started at $K = 1$ can consistently add useful components and reach better solutions than alternatives. We also examined a much bigger dataset of $N = 8.64$ million patches, and still see advantages for our MO-BM.

Finally, we perform denoising on 30 heldout images, using code from [19]. Our best MO-BM run on the $1.88$ million patch dataset achieves PSNR of $28.537$ dB, within $0.05$ dB of the PSNR achieved by [19]'s publicly-released GMM with $K = 200$ trained on a similar corpus. This performance is visually indistinguishable, highlighting the practical value of our new algorithm.

## 5 Conclusions

Our novel memoized online variational algorithm avoids noisiness and sensitivity inherent in stochastic methods. Our birth and merge moves successfully escape local optima. These innovations are applicable to common nonparametric models beyond the Dirichlet process.

**Acknowledgments** This research supported in part by ONR Award No. N00014-13-1-0644. M. Hughes supported in part by an NSF Graduate Research Fellowship under Grant No. DGE0228243.

## Footnotes

[1]To ease notation, we mark variables with hats to distinguish parameters $\hat{\theta}$ of variational factors $q$ from parameters $\theta$ of the generative model $p$. In this way, $\theta_k$ and $\hat{\theta}_k$ always have equal dimension.

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
