[Supplementary Material · MainSupplement.pdf]

# Supplementary Material: Memoized Online Variational Inference for Dirichlet Process Mixture Models

**Michael C. Hughes and Erik B. Sudderth**
Department of Computer Science, Brown University, Providence, RI 02912
mhughes@cs.brown.edu, sudderth@cs.brown.edu

## Abstract

This document contains supplementary mathematics and algorithm descriptions to help readers understand our new learning algorithm. First, in Sec. 1 we offer detailed model description and update equations for a DP-GMM with zero-mean, full-covariance Gaussian likelihood. Second, in Sec. 2 we provide step-by-step discussion of our birth move algorithm, providing a level-of-detail at which the interested reader could implement our approach.

## 1 DP mixtures with zero-mean Gaussian observations

To review, consider the generic DP mixture model defined in the main text.

$$G \sim \mathrm{DP}(\alpha_0 H), \quad G \triangleq \sum_{k=1}^{\infty} w_k \delta_{\phi_k}, \quad v_k \sim \mathrm{Beta}(1, \alpha_0), \quad w_k = v_k \prod_{\ell=1}^{k-1} (1 - v_\ell). \tag{1}$$

This process produces mixture weights $w_k$ from a *stick-breaking* process and data-generating parameters $\phi_k$ from base measure $H$. Each data item $n$ chooses an assignment $z_n \sim \mathrm{Cat}(w)$, and then draws observations $x_n \sim F(\phi_{z_n})$. We assume both $H$ and $F$ belong to exponential families

$$p(\phi_k \mid \lambda_0) = \exp\left\{ \lambda_0^T t_0(\phi_k) - a_0(\lambda_0) \right\}, \qquad p(x_n \mid \phi_k) = \exp\left\{ \phi_k^T t(x_n) - a(\phi_k) \right\}. \tag{2}$$

We now make this process concrete, providing the complete model and variational approximation for the particular case where observed data consists of a length-$D$ column vector $x_n$ and the observation model $F(x|\phi_k)$ is zero-mean Gaussian.

### 1.1 Zero-mean Gaussian Observation Model $F(x|\phi_k)$

For the Gaussian case, we parameterize the Gaussian likelihood $F(x|\phi_k)$ for component $k$ by a $D$-length mean vector $\mu_k$ and a $D \times D$ symmetric, positive definite precision matrix $\Lambda_k$. Let $\phi_k = (\mu_k, \Lambda_k)$. For the zero-mean likelihood, we assume $\mu_k = 0$ for all $k$. This leaves only precision matrix $\Lambda_k$ as a parameter of interest. The likelihood of $x_n$ when assigned to component $k$ is

$$p(x_n|z_n = k) = \mathrm{Normal}(x_n|0, \Lambda_k^{-1}) \tag{3}$$

$$\log p(x_n|z_n = k) = -\frac{D}{2} \log[2\pi] + \frac{1}{2} \log |\Lambda_k| - \frac{1}{2} x_n^T \Lambda_k x_n \tag{4}$$

$$= -\frac{D}{2} \log[2\pi] + \frac{1}{2} \log |\Lambda_k| - \frac{1}{2} \mathrm{tr}(\Lambda_k x_n x_n^T) \tag{5}$$

where $|P|$ represents the *determinant* of a square matrix $P$.

Writing the quadratic form in terms of the trace function $\mathrm{tr}(\cdot)$, which is a linear function, makes it clear that this distribution belongs to the exponential family, with sufficient statistic $t(x_n) = x_n x_n^T$. This follows from the identity $\mathrm{tr}(AB) = \mathrm{vec}(A)^T \mathrm{vec}(B)$, where $\mathrm{vec}(\cdot)$ vectorizes a $Q \times R$ matrix into a column vector of length $Q \cdot R$.

## 1.2  Wishart base measure $H(\phi_k)$

The conjugate base measure $H(\phi_k|\lambda_0)$ for this likelihood is the Wishart distribution. The parameters are $\lambda_0 = \nu, W$, where $\nu$ is a scalar degrees-of-freedom satisfying $\nu \geq D$, and $W$ is a $D \times D$ symmetric, positive definite matrix.

$$p(\Lambda_k|\nu, W) = \mathrm{Wish}(\nu, W) \tag{6}$$

$$\log p(\Lambda_k|\nu, W) = -\log \mathbb{Z}(\nu, W) + \frac{\nu - D - 1}{2} \log |\Lambda_k| - \frac{1}{2}\mathrm{tr}(W^{-1}\Lambda_k) \tag{7}$$

$$\log \mathbb{Z}(\nu, W) = \frac{\nu D}{2}\log 2 + \log \Gamma_D\left(\frac{\nu}{2}\right) - \frac{\nu}{2}\log\left|W^{-1}\right| \tag{8}$$

where $\Gamma_D(a)$ is the multivariate Gamma function, defined as $\Gamma_D(a) = \pi^{D(D-1)/4}\prod_{d=1}^{D}\Gamma(a+\frac{1-d}{2})$

## 1.3  Variational Approximation

To approximate the full (but intractable) posterior over variables $z, v, \phi$, we consider a fully-factorized variational distribution $q$, with individual factors from appropriate exponential families:

$$q(\mathbf{z}, v, \phi) = \prod_{n=1}^{N} q(z_n|\hat{r}_n) \prod_{k=1}^{K} q(v_k|\hat{\alpha}_1, \hat{\alpha}_0)q(\phi_k|\hat{\lambda}_k), \tag{9}$$

$$q(z_n) = \mathrm{Cat}(z_n \mid \hat{r}_{n1}, \ldots \hat{r}_{nK}), \quad q(v_k) = \mathrm{Beta}(v_k \mid \hat{\alpha}_{k1}, \hat{\alpha}_{k0}), \quad q(\phi_k) = H(\phi_k \mid \hat{\lambda}_k). \tag{10}$$

**Local assignments** $q(z_n)$  The posterior over assignments for each item $n - p(z_n|x_n, \phi, v) -$ is approximated by a discrete distribution over $K$ components. Although the model allows assignment to an *unbounded* set, we enforce truncation $q(z_n > K) = 0$ to make inference tractable.

Parameters $\hat{r}_{n1} \ldots \hat{r}_{nK}$ for each $q(z_n)$ must be non-negative and sum-to-one. Each $\hat{r}_{nk}$ is interpreted as the fraction of posterior responsibility that component $k$ has for $x_n$. Update equations are:

$$q(z_n) = \mathrm{Cat}(\hat{r}_{n1}, \hat{r}_{n2}, \ldots \hat{r}_{nK}) \tag{11}$$

$$\tilde{r}_{nk} = \exp\left(\mathbb{E}_q[\log w_k(v)] + \mathbb{E}_q[\log p(x_n \mid \phi_k)]\right), \quad \hat{r}_{nk} = \frac{\tilde{r}_{nk}}{\sum_{\ell=1}^{K}\tilde{r}_{n\ell}}. \tag{12}$$

Given estimates $\hat{r}_n$ for the whole dataset, we compute sufficient statistics for component $k$:

$$\hat{N}_k \triangleq \mathbb{E}_q\left[\sum_{n=1}^{N} z_{nk}\right] = \sum_{n=1}^{N}\hat{r}_{nk}, \qquad s_k(\mathbf{x}) \triangleq \mathbb{E}_q\left[\sum_{n=1}^{N}z_{nk}t(x_n)\right] = \sum_{n=1}^{N}\hat{r}_{nk}x_n x_n^T, \tag{13}$$

**Global stick-breaking parameters** $q(v)$  Each stick-breaking fraction $v_k$ is given an independent variational factor $q(v_k)$, with update equations

$$q(v_k) = \mathrm{Beta}(\hat{\alpha}_{k1}, \hat{\alpha}_{k0}), \quad \hat{\alpha}_{k1} = 1 + \hat{N}_k, \qquad \hat{\alpha}_{k0} = \alpha_0 + \sum_{\ell=k+1}^{K}\hat{N}_\ell \tag{14}$$

Given $\hat{\alpha}_{k1}, \hat{\alpha}_{k0}$ for all components, we may compute expected log mixture weights

$$\mathbb{E}_q[\log v_k] = \psi(\hat{\alpha}_{k1}) - \psi(\hat{\alpha}_{k1} + \hat{\alpha}_{k0}) \qquad \mathbb{E}_q[\log 1 - v_k] = \psi(\hat{\alpha}_{k0}) - \psi(\hat{\alpha}_{k1} + \hat{\alpha}_{k0}) \tag{15}$$

$$\mathbb{E}_q\left[\log w_k(v)\right] = \mathbb{E}_q[\log v_k] + \sum_{\ell=1}^{k-1}\mathbb{E}_q[\log 1 - v_\ell] \tag{16}$$

where $\psi(a)$ is the digamma function, the first derivative of $\log\Gamma(a)$.

**Global data-generation parameters** We define a separate factor for each component's data-generating parameters $q(\Lambda_k)$, to approximate the posterior $p(\Lambda_k|\mathbf{x}, \mathbf{z}, \ldots)$. Each factor is Wishart with parameters $\hat{\nu}_k, \hat{W}_k$, updated as follows

$$q(\Lambda_k) = \text{Wishart}(\Lambda_k|\hat{\nu}_k, \hat{W}_k) \tag{17}$$

$$\hat{\nu}_k = \nu + \hat{N}_k, \qquad \hat{W}_k^{-1} = W^{-1} + s_k(\mathbf{x}) \tag{18}$$

Given $\hat{\nu}_k, \hat{W}_k^{-1}$, we compute the expected log probability under component $k$ for each data item $x_n$

$$\mathbb{E}_q\Big[\log p(x_n|\phi_k)\Big] = -\frac{D}{2}\log[2\pi] + \frac{1}{2}\mathbb{E}_q\Big[\log|\Lambda_k|\Big] - \frac{1}{2}\text{tr}(\mathbb{E}_q[\Lambda_k]x_n x_n^T) \tag{19}$$

Here, we use basic expectations under the Wishart distribution:

$$\mathbb{E}_q[\Lambda_k] = \hat{\nu}_k \hat{W}_k, \qquad \mathbb{E}_q[\log|\Lambda_k|] = \psi_D\Big(\frac{\hat{\nu}_k}{2}\Big) + D\log 2 + \log|\hat{W}_k| \tag{20}$$

where $\psi_D(a) = \sum_{d=1}^{D} \psi(a + \frac{1-d}{2})$ is the multivariate digamma function of dimension $D$.

## 2 Birth Moves for Mixture Models

**Overview.** As input, our birth procedure takes an existing variational model $q$ with $K$ components, together with global sufficient statistics $S^0 = [S_1^0\ S_2^0\ \ldots\ S_K^0]$ for the full dataset $\mathbf{x}$. The algorithm consists of 3 steps: *collection* of a subsample dataset $\mathbf{x}'$, *creation* of brand-new components by a fresh DP mixture model variational analysis of $\mathbf{x}'$, and *adoption* of these fresh new components by the full dataset $\mathbf{x}$. The output will be an expanded model $q^*$ with $K + J'$ components.

### 2.1 Collection of the target dataset $\mathbf{x}'$

We find it simplest to focus on a birth move which *targets* a specific component $k'$. After selecting the component $k'$, the birth move proceeds to subsample data $\mathbf{x}'$ associated with $k'$, using the existing local assignment factors $q(z_n)$ to identify which data items to subsample. Certainly other ways of subsampling exist, but this has an intuitive interpretation as targeting a single sub-optimal component which may be too coarse (explaining multiple ideal subclusters) and refining it.

**Selecting the target component $k'$.** The procedure for selecting which component $k'$ to target is not complicated. For understanding the mechanics of birth moves, it is fine to simply select the component $k'$ uniformly at random. If we have $K$ active components in original model $q$, then

$$k' \sim \text{Unif}(\{1, 2, \ldots K\}) \tag{21}$$

Many other schemes for choosing $k'$ can be considered. But the above is perfectly sufficient, albeit potentially slow at trying a diverse set of possible moves in a short timespan.

In practice, we recommend sampling $k'$ at random, but in a way that *biases* towards choosing components that (1) have more mass and (2) have not been targeted in the last few moves. Let $\hat{N}_k^0$ give the current expected count on the full dataset, and $L_k$ denote the number of passes through the data since component $k$ was last chosen for a birth move.

$$p(k' = k) \propto (\hat{N}_k^0) * (L_k)^2 \tag{22}$$

Squaring the $L_k$ term forces the algorithm to not wait very long between trying all possible components, ensuring good coverage of the space of all possible moves. We found that this revised selection procedure improved the speed with which our algorithm recovered all missing components, but uniform selection should eventually reach the same high-quality configurations.

**Sampling a dataset targeted on component $k'$.** After selecting $k'$, next we collect a *targeted* dataset $\mathbf{x}'$ with size at most $N'$. We recommend choosing $N'$ large enough that necessary "undiscovered" components (not in the existing set $\{1, 2, \ldots K\}$ can be learned, but still small enough that running many batch VB iterations does not take more than a few seconds. We found $N' = 10000$

to be a good choice for our experiments using Gaussian likelihoods with dimension $D = 25$ to $D = 50$. For small values like $D = 2$, $N'$ in the low hundreds may be sufficient.

The target dataset $\mathbf{x}'$ contains samples without replacement from the full dataset $\mathbf{x}$ (of size $N$). For each observed vector $x_n \in \mathbf{x}$, we add it to our subsample $\mathbf{x}'$ if the following test is true:

$$\hat{r}_{nk'} > \tau, \quad \text{with typical value } \tau = 0.1 \tag{23}$$

Here, $\hat{r}_{nk'}$ is interpreted as the posterior responsibility of component $k'$ for data item $n$. Each observation $n$ has a vector $[\hat{r}_{n1} \ \hat{r}_{n2} \ \cdots \ \hat{r}_{nK}]$ of these responsibilities, where each entry is non-negative and the whole vector sums to one. The value $\hat{r}_{nk'} \in [0, 1]$ will be larger than the threshold $\tau$ if the $n$-th observation is well-explained by component $k'$.

Intuitively, our simple "threshold" test for adding data to the targeted dataset $\mathbf{x}'$ ensures that the subsample contains data which are significantly explained by component $k'$, while also promoting diversity (since members could also be partially explained by some other component). The threshold of $0.1$ strikes a good balance between these competing goals. We did explore a few other values for $\tau$ among $\{0.2, 0.5\}$ in preliminary experiments, and found that $\tau = 0.1$ performed slightly better. We stress that this does not need to be fine-tuned for the particular dataset at hand: the same setting was used for all our experiments.

In practice collection is done by visiting each batch in turn, and collecting all relevant data items until the size of $\mathbf{x}'$ exceeds the limit $N'$. When batch traversal order is randomized at each pass through the data, this has the beneficial effect of randomizing the subsample.

## 2.2 Creating an expanded model with brand-new components from the targeted dataset

Next, we consider adding new components to our existing model. We first train a fresh DP mixture model with $K'$ brand-new components on $\mathbf{x}'$ via conventional (batch) variational inference, and then later combine these components with the existing $K$ component model.

The process of creating components by a fresh variational analysis is general and elegant. This strategy applies to *any* DP mixture with exponential family likelihoods, re-uses existing code routines needed for the larger learning algorithm, and has a pleasing interpretation as a "divide-and-conquer" strategy. That is, to find the ideal clustering for the large dataset $\mathbf{x}$, we simply need to repeatedly find some broadly related subset $\mathbf{x}'$ and perform a more fine-grained clustering of that subset.

**Creation of new components.** Given the target dataset $\mathbf{x}'$ as a stand-alone dataset for analysis, we perform one run of standard full-dataset variational inference. We fit a $K'$-component DP mixture model with exactly the same prior parameters as the original model.

In practice, we initialize by setting fixed-truncation $K' = 10$, which is a reasonable compromise between diversity and speed. To initialize, we select $K'$ observations (uniformly at random) from $\mathbf{x}'$ to seed parameters. We run only for a fixed budget of $I' = 100$ iterations or until convergence of the objective, whichever happens first.

The choices of truncation level $K'$, initialization routine, and number of iterations $I'$ may all impact the performance of the birth move. We found the same settings lead to reasonable performance across all tested datasets. In general, a more intelligent initialization is better. Running for longer will produce more refined components, but at the cost of increased run-time.

After the run, instead of saving estimated parameters we save *summaries* for each new component:

$$\hat{N}' = [\hat{N}_1 \ \hat{N}_2 \ \cdots \ \hat{N}_{K'}] \tag{24}$$

$$s(\mathbf{x}') = [s_1(\mathbf{x}') \ s_2(\mathbf{x}') \ \cdots \ s_{K'}(\mathbf{x}')] \tag{25}$$

In general, some final components may have very few assignments to data $\mathbf{x}'$. Some may be empty or nearly-empty. We thus post-process results to remove components $j$ which have low expected counts $\hat{N}_j$ for explaining the data $\mathbf{x}'$. Pruning out empty components makes later phases much faster without sacrificing quality.

Specifically, we remove component $j$ if $\hat{N}_j < \epsilon N'$, and we set $\epsilon = \frac{1}{20}$. After this removal, we end up with a set of $J'$ sufficient statistics $\{\hat{N}_j, s_j(\mathbf{x}')\}_{j=1}^{J'}$, where $J' \leq K'$. These sufficient statistics are all we pass along to the next step.

If only $J' = 1$ component is left, by construction its summary will be very close to the summary for the target component $k'$. We shouldn't expect adding this new component will improve the original model (since $k'$ already exists unchanged). Thus, if the resulting number of components is $J' = 1$, we abort the birth process early and return to the original $K$ component model.

**Creation of combined model**  Here, we combine the $K$ components from the existing model with the brand-new $J'$ components. Working purely in terms of sufficient statistics, we find that it is easy to build a coherent combined model simply by *concatenating* the fresh components $S' = [\hat{N}' \ s(\mathbf{x}')]$ onto the existing global sufficient statistics $S^0 = [\hat{N} \ s(\mathbf{x})]$.

We now have an expanded model with $K + J'$ summaries, $S^* = [\hat{N}^* s^*]$:

$$\hat{N}^* = [\hat{N}_1 \ \hat{N}_2 \ \cdots \ \hat{N}_K \ \hat{N}'_{K+1} \ \hat{N}'_{K+2} \ \cdots \ \hat{N}'_{K+J'}] \tag{26}$$

$$s^* = [s_1(\mathbf{x}) \ s_2(\mathbf{x}) \ \cdots \ s_K(\mathbf{x}) \ s_{K+1}(\mathbf{x}') \ s_{K+2}(\mathbf{x}') \ \cdots \ s_{K+J'}(\mathbf{x}')] \tag{27}$$

This concatenation creates a valid set of sufficient statistics for an "expanded" dataset formed by the union of $\mathbf{x}$ and $\mathbf{x}'$. This set "double-counts" the subsample $\mathbf{x}'$, assigning these data items to both original components (mostly $k'$) and new components $K + 1, \ldots K + J'$. In the next phase (adoption), we pass through the entire dataset, and discover which interpretation (original or new components) is preferred by the model.

Using new, expanded sufficient statistics $S^*$, we can then expand both local and global factors. The resulting expanded model $q^*$ remains valid due to our *nested* truncation of the variational posterior. At this stage, no local parameters have been assigned to the new components. For all $n$, we simply expand $q^*(z_n)$ to be a discrete distribution over $K + J'$ components, where only the first $K$ have mass:

$$\text{Before: } q(z_n) = \text{Cat}(\hat{r}_{n1}, \ldots \hat{r}_{nK}) \tag{28}$$

$$\text{After: } q^*(z_n) = \text{Cat}(\hat{r}_{n1}, \ldots \hat{r}_{nK}, 0, 0, \ldots 0) \tag{29}$$

Crucially, all parameters $\hat{r}_{nk}$ are directly transfered from the previous model $q$, and no batches actually need to be visited at this stage (they can instead be lazily expanded during each visit of the adoption pass). Another consequence of this construction is that $q^*(\phi_k) = q(\phi_k)$ for all original components $k = 1, 2, \ldots K$, including the target component $k'$. For new components $j$, $q^*(\phi_j)$ are set to the resulting factors from the targeted analysis.

Only the stick-breaking factors $q^*(v)$ must be completely re-written after the expansion. Expansion forces these factors to shift probability mass onto newly inserted components. Given the counts from all $K + J'$ summaries $S^*$, the update equations become

$$q^*(v_k)|S^* = \text{Beta}(v_k|\alpha^*_{k1}, \alpha^*_{k0}), \qquad \hat{\alpha}^*_{k1} = 1 + \hat{N}_k, \quad \hat{\alpha}^*_{k0} = \alpha_0 + \sum_{\ell=k+1}^{K+J'} \hat{N}_k \tag{30}$$

The choice to insert new components last in the stick-breaking order (which is implicitly done by concatenation) is fairly principled. On average, freshly discovered components will be more "rare" than the original ones, and so will likely have smaller effective mass. Since the stick-breaking construction is "size-biased", inserting components with smaller mass later in the order makes sense.

## 2.3  Adoption of the new components

After expanding the model to have $K + J'$ components, we then proceed normally through the memoized variational inference E-step (local factors) and M-step (global factors) at each batch. Our goal in this pass is to have newborn components become "adopted" by the original dataset $\mathbf{x}$, attaining critical mass by actively explaining some data in $\mathbf{x}$.

At the start of this pass, we have the expanded set of global sufficient statistics $S^*$ described earlier. Retaining the target summaries $S'$ as well as the previous global summaries $S^0$ within $S^*$ allows each brand-new component a chance to influence several batches of data.

To understand this necessity, consider the alternative: after creating an expanded model, we discard $S'$ and keep only the original summaries $S^0$. To be consistent with local assignments, we must

expand $S^0$ to have zero mass on new components $K+1, K+2, \ldots K+J'$. Now, imagine visiting the first batch of data $\mathcal{B}_1$ and performing the E-step. After this step, the only mass on new components will come from the current batch. For each new component $j$: $S_j^0 = S_j(\mathcal{B}_1)$. If this batch does not assign any mass to component $j$, then $s_j^0 = 0$. Next, the following M-step (see the main text's Eq. (9)) will completely rewrite $\hat{\lambda}_j = \lambda_0 + 0 = \lambda_0$, reseting to its prior value. Component $j$ will lose all information from the targeted dataset $\mathbf{x}'$ after only one update, becoming useless even though later batches may have highly preferred it.

To avoid this disaster, we choose to retain the "dual" interpretations of the data $\mathbf{x}'$ in $S^*$ throughout the pass, which ensures every brand-new component $j$ always has mass at least $\hat{N}_j'$. Thus, even when the first batch is not assigned at all to component $j$, we'll have $s_j^* = s_j(\mathbf{x}')$, and the update $\hat{\lambda}_j = \lambda_0 + s_j(\mathbf{x}')$ will retain vital information from our targeted analysis.

At the end of the adoption pass, immediately before the last M-step update to global parameters, we subtract-away all targeted summaries $S'$ from the final global summaries $S^*$. This ensures that by the end of the adoption pass, both the final global summary and all global factors $q^*(v), q^*(\phi)$ have scale exactly consistent with the dataset $\mathbf{x}$. Under these conditions, the ELBO can be calculated exactly and merges can proceed.

## 2.4  Multiple birth moves in one pass

As a final note, performing several birth moves during one pass (refining multiple components at once) is definitely possible. We need only to collect several subsampled datasets $\mathbf{x}_1', \mathbf{x}_2', \ldots$, discover new components from each one via separate variational analyses, and then adopt all new components into an expanded model. For simplicity we focus on just one birth in the description below. All our experiments perform just one birth per pass, except for the final analysis of $8 \times 8$ image patches, where we execute two births per pass.