[Reviews · NeurIPS 2013]

Submitted by Assigned_Reviewer_1

The authors proposed a novel online variational inference, called memoized online variational inference (MOV inference).
The proposed online algorithm is not technically new.
It is not a stochastic gradient (or step-wise) type online algorithm but an incremental algorithm which updates parameters by replacing old statistics with new one.
This framework is proposed by Neal & Hinton.
The difference is trivial because Neal & Hinton incremental algorithm is based on the EM algorithm and updates parameters for each data point, while the proposed inference is based on the variational inference and updates parameters for each batch.

The authors proposed a novel framework for adapting the number of topics, called birth and merge method.
The existing construction of the new components is either heuristic or highly specialized to the particular model.
The proposed birth moves can add many new components at once, and are applicable to any exponential family mixture model.
This framework is interesting and a strong point.
However, even if the existing work is either heuristic or highly specialized to the particular model, you should compare the existing methods with the proposed method such as [4] and [12].
In particular, [4] and [12] can be applicable for the Gaussian mixture model.



Summary: The birth and merge algorithm is interesting.
However, the experimental setting is not convincing.
The stochastic variational inference is not state-of-the-art in the truncation-free setting.

Submitted by Assigned_Reviewer_5

The authors propose a novel method for performing memory efficient inference in Dirichlet Process (DP) models. There are two key advances presented in this work: memoized online (MO) learning of the model parameters using small subsets of data; an efficient birth-merge move for dynamically altering the truncation level of the stick breaking approximation.

The first advance is quite similar to approaches for memoized learning with EM algorithms. To my knowledge this approach has not been applied to non-paramteric Bayesian models however, which is the major novelty of this part of the work.

The birth merge move is in my view the more interesting advance in this paper, as it elegantly handles the issue of proposing births while keeping a limited subset of the data in memory. In the results section the authors apply the model to several synthetic and real datasets and show that MO learning is able to escape local optima, particularly when used with the birth merge move.

The paper is generally well written and easy to follow. There are some small points (listed below) which would make the exposition easier to follow and the experiments reproducible. In particular the specific settings for the birth merge parameters would be useful. The work is a nice synthesis of previous work in different settings along with some novel contributions. The problem of scaling DPs to large datasets is generally important since it allows for clustering with automatic inference of model complexity. The authors method is simple to implement, efficient and (except for births or merges) maintains the monotonic increase of the lower bound.

Minor Revisions:

- The exposition of the memoized batch updates is excellent and can be readily implemented from the authors description. One thing which would make it even easier would be if the authors were to provide the complete set of update equations for the Gaussian emission used in the experiments in a supplemental file. This would be particularly helpful to readers not familiar with VB methods, for example the computation of $E_{q}[log w_{k} (v)]$.

- When trying to implement the birth/merge moves I found the following points could be clarified in the text or a supplemental file with pseudo code:

1) How was $N'$ (the size of the birth batches) set in the experiments.

2) How was the DP fit in the creation step of the birth move. That is was Gibbs, batch VB, etc. used. I assume batch VB was used but it not stated explicitly as far a I can see.

3) How exactly are the sufficient statistics inflated to protect newborn clusters in the third phase of the birth proposals.

4) For the merge moves clarify what exactly is cached to compute the entropy term. Is it a single set O(K^2) values or one set of values per batch.

- The authors mention where components are inserted and deleted in the size biased ordering of the stick lengths several times. Could they clarify the significance of these choices.

- I am not sure why the authors did not compare against stochastic online methods using adaptive learning rates, which is in reference [2] from the paper.

- I found the bottom row of figure 2 somewhat confusing. What do the colours used in the covariance matrix indicate?

- It would be great if the code for this method and the other methods used in the comparisons could be made available.
Summary: A good paper which is worthy of publication after some small revisions.

Submitted by Assigned_Reviewer_6

The proposed "memoized online variational inference" is a natural technique for exploiting the additivity of sufficient statistics for a certain class of exponential family probabilistic models. At it's core, the method is simply propagating information between the global and local latent random variables more frequently than traditional VB. In some of my previous research I've actually used this technique for a specific model, and assumed it was a standard trick for VB (& not knowing of the authors' research), and found that it greatly improved my results. So I can independently attest that memoized updates should be highly applicable/beneficial to VB practitioners. I'm glad to see a paper explicitly work out the updates.

The birth/merge updates are neat, but they're still heuristics: on line 216 the authors mention that "construction of these new components is either heuristic [4] or highly specialized...In contrast our birth moves..." when referring to previous work (but their method is entirely heuristic as well). Other than that, these heuristics seem well-thought-out, and I plan to give them a shot in some of my own work.

How exactly does the citation on line 202 (to [11]: A stochastic gradient method with an exponential convergence rate for finite training sets) motivate your use of cached gradients?

The experiments seem solid, but I encourage the authors to release their experimental code. Of course, the conclusions drawn in this section are data/model dependent, but the authors make an above-average effort to understand their algorithm and test on a number of different circumstances/data. I appreciate that the authors separately compared the memoized algorithm with and without the birth/merge steps.

I would have liked to see a different model besides the DP-GMM explored: the memoized result applies to general exponential family models, after-all. From Figure 2: why does the MO and MO-BM results not depend on the number of batches? Are the results the same with N batches? I would like to have seen an analysis of batch size and its effect on the inference results.

The "discussion" section is a misnomer, perhaps use "summary."

NB: The experimental line plots are not discernible when printing from a black and white printer.


Review summary:

The quality of this paper is high: the results are technically sound and the claims are well supported by their diverse experimental section.

The paper is clearly written and organized.

I would like to run some of these experiments myself, and while the experimental section is fairly detailed, a link to the actual experimental code would be helpful.

The paper is fairly original. My initial perspective was that their update results were somewhat trivial, but a quick literature search indicates that their update method is not (explicitly) in widespread use. It's certainly beneficial to share these types of helpful tricks to practitioners in our field.
Summary: This paper is a solid contribution in VB inference that applies to a fairly broad class of probabilistic models.
Author Feedback

Author rebuttal: We thank the reviewers for their helpful comments. In our paper, we develop memoized online variational bayes (MO-VB), an inference algorithm that does not require any learning rate and is insensitive to batch size. Compared to alternative methods like stochastic online variational (SO-VB), our approach delivers much more consistent performance without the need for extensive parameter tuning. We would like to emphasize that while our approach builds on a rich variational inference literature, it differs in several key respects which are crucial to its scalability, robustness to initialization (via birth-merge moves), and applicability to nonparametric models.

NOVELTY VS. NEIL & HINTON (R1):
In some sense, all modern work on variational methods follows closely from the ideas in Neal & Hinton (and various tutorials by Jordan and colleagues, and decades-older work in statistical physics). But there are several novel contributions which do not follow easily, including:
* The nested family of truncations which allows these ideas to be correctly applied to Bayesian nonparametric models.
* The practical demonstration that compared to SO-VB, the MO-VB approach is simpler (in implementation and parameter tuning), equally scalable, and often more accurate.
* The development of birth-merge moves which operate on batches of data, but nevertheless avoid local optima and (in the case of merges) are guaranteed to increase the full-data likelihood bound.
* Achieving all of this in a framework that is directly applicable to a broad range of models.

CODE (R5 & R6):
We are pleased that the reviewers show interest in the code for our algorithms. We are in fact working on an open source, Python software package which implements the various inference frameworks compared in this paper for a range of Bayesian nonparametric models (mixture models, topic models, relational models, time series models, etc.). This code will be publicly released.

EXPERIMENTAL COMPARISONS (R1 & R5):
* Bryant & Sudderth, NIPS 2012 [4]: In line 347, we do compare our merge moves (which have provable guarantees to improve the ELBO) to the merges found in [4], which use only the current minibatch and have no whole-dataset guarantees. We find that [4]'s merges can cause the ELBO to drop significantly, as shown in Fig. 2's "GreedyMerge" traces.
* Wang & Blei, NIPS 2012 [12]: This work relies on a local collapsed Gibbs sampler to create new components. Such samplers are widely known to be slow to add needed components (and become less effective as dataset size grows). In some internal experiments, we replaced the batch VB in the creation step of the birth move with a collapsed gibbs sampler, and found that speed was slower and performance worse (sometimes slightly, sometimes substantially). We excluded these comparisons due to space constraints.
* Ranganath et al., ICML 2013 [2]: Code is not available for the SO-VB adaptive learning rates proposed in this paper, which became available just before the NIPS deadline. We found the details of their approach too complex to reproduce. Our MO-VB lacks learning rates entirely, making it much simpler.

DETAILED COMMENTS:
R6: why do MO-VB results not depend on batch size (Fig. 2)?
MO-VB global parameter updates always account for the whole dataset via memoized sufficient statistics. This makes our algorithm very insensitive to batch size, since the influence of each batch is balanced at every update. In contrast, SO-VB is much more sensitive due to its reliance on the current batch alone.

R6: new birth moves are well-thought-out, but still should be called "heuristic".
We'll find a better word to describe the differences. Our method is certainly more general-purpose than prior work: creating new components by running batchVB on a targeted subsampling of data is applicable to most popular models.

R5: some algorithm details missing.
We appreciate the detailed list. We'll improve figure captions and add needed information to a supplement or longer technical report.

R6: would have liked comparison of MO-VB on another model in addition to GMM.
We are in fact working on extensions to other models, but even fully describing our work on the GMM was difficult in the NIPS page constraints. We intend to experiment more broadly in future papers.

R6: connection to ref [11] is unclear.
We'll improve the description. Like our MO-VB approach, [11]'s algorithm exploits additivity to store a memoized version of a "whole-dataset" gradient that can be updated incrementally.